# Influence of Long-Term Freezing of Carcasses in Pre- and Post-Rigor Mortis Stages on the Technological and Nutritional Parameters of the *Longissimus lumborum* Muscle of Botucatu Rabbits

**DOI:** 10.3390/ani14172510

**Published:** 2024-08-29

**Authors:** Daniel Rodrigues Dutra, Erick Alonso Villegas-Cayllahua, Giovanna Garcia Baptista, Lucas Emannuel Ferreira, Érika Nayara Freire Cavalcanti, Nívea Maria Gomes Misson Carneiro, Ana Veronica Lino Dias, Mainara Carolina Francelino, Mateus Roberto Pereira, Leandro Dalcin Castilha, Hirasilva Borba

**Affiliations:** 1São Paulo State University (UNESP), Faculty of Agricultural and Veterinary Sciences, Campus Jaboticabal, Jaboticabal 14884-900, Brazil; 2Department of Animal Science, State University of Maringá, Maringá 87020-900, Brazil

**Keywords:** meat quality, meat science, *Oryctolagus cuniculus*, rabbit meat, storage, tenderness

## Abstract

**Simple Summary:**

Comparisons of meat quality in pre- and post-rigor mortis phases have been the focus of studies in conventional species, but little is known about the quality of pre-rigor mortis rabbit meat, particularly during long-term storage. This study investigated how freezing rabbit meat before and after rigor mortis affects its quality over a 12-month period. Rigor mortis, characterized by post-mortem muscle stiffening, influences meat texture and quality. The results showed that fresh meat frozen before rigor mortis (pre-rigor) was softer, moister, and less acidic compared to that frozen after rigor mortis (post-rigor), which exhibited a higher level of redness. Over time, all of the meat samples showed reduced redness, increased yellowness, a higher acidity, and a loss of moisture and minerals. Despite these changes, freezing effectively improved meat tenderness and preserved its physical, chemical, and nutritional quality for up to 12 months. This aligns with recommendations from the United States Department of Agriculture, which states that rabbit carcasses can be frozen for up to 12 months without compromising quality. These findings provide valuable insights for the rabbit meat industry, suggesting that freezing, either in the pre-rigor or post-rigor phase, is an effective method to maintain meat quality and deliver high-quality products to consumers.

**Abstract:**

The aim was to assess the impact of long-term storage on the quality of Botucatu rabbit meat frozen in pre- and post-rigor stages. The stability of the technological and nutritional parameters of *Longissimus lumborum* (LL) muscle was analyzed over 12 months. In the post-rigor phase, the dorsal LL surface showed a higher level (*p* < 0.05) of redness and saturation, while the ventral surface showed a higher level (*p* < 0.05) of yellowness compared to the pre-rigor LL muscle. During storage, the redness and saturation in the LL muscle decreased (*p* < 0.05), while the yellowness increased (*p* < 0.05) on both dorsal and ventral surfaces. In the first six months, the pre-rigor meat had a higher pH (*p* < 0.05) compared to the post-rigor meat. The fresh meat showed higher (*p* < 0.05) shear force values in the post-rigor stage. Over the 12-month study period, the lipid oxidation, myofibrillar fragmentation index, gross energy, and levels of protein, fat, and carbohydrates increased (*p* < 0.05), while the shear force, mineral content, and moisture decreased (*p* < 0.05). Thus, rigor mortis affects meat color in Botucatu rabbits. Fresh meat in the pre-rigor stage is softer, moister, and less acidic than post-rigor meat after 24 h of chilling. Long-term freezing enhances tenderness, regardless of the rigor phase at freezing, preserving its physical, chemical, and nutritional quality, with minor changes in color, lipid oxidation, and chemical composition.

## 1. Introduction

The rabbit meat industry has been striving for the greater technological advancement and standardization of its processes related to the cold chain, aiming to offer consumers a high-quality product, to become increasingly competitive in the global meat market, and to focus on the production and commercialization of whole carcasses and frozen cuts. Freezing ensures the maintenance of meat product quality over long storage periods, as the cold preserves muscle tissue structures and slows down the biochemical reactions responsible for food deterioration. It causes minimal changes to its physicochemical characteristics, such as protein denaturation, meat discoloration, and increased lipid oxidation, when compared to the changes seen in fresh meat [1]. The United States Department of Agriculture [2] itself states that rabbit carcasses can be frozen for up to 12 months without compromising quality.

However, to reduce costs associated with electricity, labor, and losses from cooling and dripping due to the prolonged operation of cold storage rooms [3,4], many rabbit slaughterhouses in various Latin American countries, including Brazil, have prioritized the commercialization of frozen carcasses without prior chilling. In this process, hot carcasses are immediately stored in rapid freezing tunnels at temperatures below −18 °C before rigor mortis sets in, leading to tougher meat due to a process known as thaw rigor [5,6,7]. In contrast, the high pH of pre-rigor carcasses could lead to rabbit meat with a higher water-holding capacity and lower level of cooking loss, which are considered to be parameters of great importance in raw meat processing, as observed in some domestic species [8,9,10].

Although the comparison of meat quality in pre- and post-rigor periods has been the subject of studies for other species, such as pigs [11], chickens [12], turkeys [9], lambs [10], and cattle [4], little is known about the quality of pre-rigor rabbit meat, especially its long-term storage. Additionally, Asian countries such as China, the world’s largest producer of rabbit meat with an annual production of 358,152 tons [13], represent an important market niche to be explored, as pre-rigor meat is highly appreciated by the population and is part of local consumption habits, having been commercialized for decades [14,15].

Given this scenario, the present study aimed to evaluate the effect of prolonged storage on the technological and nutritional quality losses of rabbit meat frozen before and after the establishment of rigor mortis by assessing changes in the physicochemical parameters and chemical composition of *Longissimus lumborum* muscle over a period of up to 12 months of frozen storage.

## 2. Materials and Methods

### 2.1. Animals, Slaughter, and Carcasses

A total of 60 Botucatu male rabbits [16,17], weaned at 35 days of age, were randomly selected for this study. The animals were individually housed and raised in flat-deck cages under the same ambient conditions, feeding regimens, and sanitary management until they were slaughtered (90 days) at the Rabbitry Sector of the Faculty of Agricultural and Veterinary Sciences of São Paulo State University (FCAV/UNESP), Jaboticabal Campus, SP, Brazil (21°14′ S, 48°17′ W, 583 m altitude).

During this period, the rabbits were fed coast-cross grass hay (*Cynodon dactylon* (L.) Pers.) ad libitum, along with a controlled amount of pelleted standard feed. The mixed ration contained 14% crude protein, 3% fat, 18% crude fiber, 15% mineral matter, 5% phosphorus, 10% calcium, and 13% moisture, meeting the nutritional requirements for the animal category, according to De Blas and Wiseman [18].

The rabbits were slaughtered at an average live weight of 3.1 kg, with approval from the Institutional Animal Ethics Committee (CEUA) of the aforementioned institution (protocol no. 1924/22), and subjected to a 12 h fasting period and a 4 h pre-slaughter rest period [19]. The entire slaughtering process complied with the operational requirements of the Regulation for Industrial and Sanitary Inspection of Animal Products [20] and Regulation (EC) Nº. 1099/2009 [21].

After evisceration, a group of 30 hot carcasses were stored in a climatic chamber at a temperature of 2 ± 2 °C for 24 h to ensure complete establishment and resolution of rigor mortis. Once chilled, 20 cold carcasses in the post-rigor stage were stored in a rapid freezing tunnel at −20 ± 2 °C, while the remaining 10 cold carcasses in the post-rigor stage were separated and sent under refrigeration (±4 °C) to the Food Analysis Laboratory (LaOra) of FCAV/UNESP for physicochemical analyses.

A second group of hot carcasses (n = 30) was not subjected to the chilling process after evisceration. Among them, 20 hot carcasses in the pre-rigor stage were immediately frozen at −20 ± 2 °C, while the remaining 10 hot carcasses were sequentially subjected to respective laboratory analyses while still in the pre-rigor stage.

The frozen carcasses were individually vacuum-packed in polyethylene plastic bags and stored at −20 ± 0.5 °C for up to 12 months. At the end of each pre-established time period (6 and 12 months), the carcasses were slowly thawed under refrigeration at 4 °C for 24 h in a BOD incubator (Eletrolab EL101/3 116 250W, Eletrolab, São Paulo, SP, Brazil). The carcasses were randomly selected to be placed in the chilling chamber and then randomly assigned to the following six groups, based on the cooling process adopted:NC0: Meat in pre-rigor stage without freezing;NC6: Meat in pre-rigor stage frozen for 6 months;NC12: Meat in pre-rigor stage frozen for 12 months;C0: Meat in post-rigor stage without freezing;C6: Meat in post-rigor stage frozen for 6 months;C12: Meat in post-rigor stage frozen for 12 months*NC = without 24 h chilling; C = with 24 h chilling.

Each group consisted of ten carcasses. At the end of each designated time period, the *Longissimus lumborum* muscle was excised from the right and left sides of the carcasses for evaluation of physicochemical properties.

### 2.2. Analysis of Technological Parameters

#### 2.2.1. Color

A Minolta CR-400 colorimeter (Konica Minolta Sensing, Inc., Osaka, Japan) (with the following settings: diffuse lighting/0 viewing angle, D65 illuminant, specular component included, with 8 mm aperture size) and the CIELAB system (*L**, *a**, and *b**) were used [22]. Luminosity (*L**), red intensity (*a**), and yellow intensity (*b**) were evaluated across the entire length of the LL muscle excised from the left side of the carcass.

Meat color was assessed in triplicate, immediately after 30 min of exposure to air, on both the dorsal and ventral surfaces of the muscle. Subsequently, the saturation index (Chroma—*C**) was calculated as follows, as demonstrated by Ramos and Gomide [23]:*C** = (*a**^2^ + *b**^2^)1/2

#### 2.2.2. pH

The pH was determined in triplicate using a digital pH meter (Testo 205, Testo Inc., Sparta, NJ, USA) coupled with a penetration electrode, inserted into the LL muscle at the level of the 5th lumbar vertebra on the right side of the carcass [24]. The temperature compensation was automatically adjusted by the device, which had been previously calibrated using buffer solutions with pH values of 4.00 ± 0.02 and 7.00 ± 0.02.

#### 2.2.3. Water-Holding Capacity (WHC)

WHC was determined in triplicate following the methodology described by Hamm [25]. Two grams of sample from the caudal portion of the left side LL muscle of the carcasses were placed between two filter papers and acrylic plates and subjected to pressure exerted by a 10 kg weight for five minutes. Subsequently, the samples were reweighed to determine the WHC, expressed as a percentage.

#### 2.2.4. Cooking Loss (CL)

CL was determined from samples collected from the caudal portion of the left side of the LL muscle of the carcasses, using the methodology described by Honikel [26]. Samples of approximately equal size and weight were weighed, individually packed, and cooked in a water bath (85 °C) for 40 min in one batch only. After being cooled to room temperature, they were reweighed to determine the cooking loss, expressed as a percentage.

#### 2.2.5. Shear Force (SF)

Cooked samples from the CL analysis were used, which were cut into approximately 1 cm^2^ sections, positioned with fibers oriented perpendicular to the Warner–Bratzler device (attached to the Texture Analyser TA-XT2i, Stable Micro Systems Ltd., Godalming, UK), and subjected to shear force tests in triplicate. The force required to shear the samples was expressed in Newtons, following the method described by Lyon et al. [27].

#### 2.2.6. Myofibrillar Fragmentation Index (MFI)

MFI was determined in triplicate using the methodology proposed by Culler et al. [28], complemented with the use of the biuret technique [29] to determine protein concentration in the myofibril suspension derived from samples collected by scraping the cranial portion of the right side of the LL muscle of the carcass, using the following formula:MFI = optical density × 200.

#### 2.2.7. Sarcomere Length

Sarcomere length was determined using phase-contrast microscopy [30]. Subsamples weighing 0.5 g were homogenized from the cranial portion of the right side of the LL muscle of the carcass using an Ultra-Turrax homogenizer (Marconi MA102, Marconi Equipamentos Para Laboratórios Ltda., Piracicaba, São Paulo, Brazil) with 30 mL of a mixed solution containing 0.08 mol/L potassium chloride and 0.08 mol/L potassium iodide at a speed of 15.000 rpm for 30 s to disrupt cells and facilitate removal of myofibrils into suspension.

A drop of the homogenate was transferred to a slide and covered with a coverslip. Individual slides were prepared for each slaughtered animal, with a minimum of 12 readings per slide using phase contrast on an optical microscope (Novel BM2100, Nanjing Jlangnan Novel Optics., Ltd., Nanjing, China) at 1000× magnification (100× objective, 10× ocular), using oil immersion on the coverslip. Sarcomere length is expressed in µm.

#### 2.2.8. Lipid Oxidation

Five grams of ground samples from the cranial portion of the right side of the LL muscle of the carcass were collected for extraction using trichloroacetic acid and determined in triplicate using the thiobarbituric acid reactive substances (TBARS) test, following the methodology described by Vyncke et al. [31]. After coloration with thiobarbituric acid, readings were taken at a wavelength of 532 nm, with results expressed in mg of malondialdehyde (MDA) per kg of sample.

### 2.3. Analysis of Nutritional Parameters

#### 2.3.1. Chemical Composition and Soluble Protein

Samples of the LL were excised from the right side of the carcass and lyophilized for 72 h (Super Modulyo 220, Thermo Fisher Scientific Inc., Waltham, MA, USA) to determine moisture content (method 950.46), protein content (method 977.14), and ash content (method 920.153) according to the methods recommended by the Association of Official Analytical Chemists [32]. Total lipid content [33] and carbohydrate content [34] were also determined. Gross energy was determined according to Merril and Watt [35]. The exudate from the carcasses was collected after 24 h of thawing at the end of each storage period to determine soluble protein using the method described by Hartree [36].

#### 2.3.2. Cholesterol

Cholesterol was quantified using the cold technique proposed by Folch et al. [37], which was adapted by Saldanha et al. [38] and subsequently by Oliveira et al. [39]. A 0.5 g sample of LL excised from the right side of the carcass was weighed and lyophilized in a 50 mL Falcon tube, to which 6 mL of ethanol and 4 mL of 50% KOH were added. The tubes were kept in a water bath with agitation, at 40 °C, until the samples were completely dissolved, then they were maintained at 60 °C for an additional 10 min. Ten milliliters of N-hexane were added, and part of the solution was removed to proceed with phase separation, with this step being repeated three times.

Aliquots of 3 mL were pipetted from the upper phase into test tubes and dried using nitrogen gas. After drying, 0.5 mL of isopropyl alcohol was added to each tube. The tubes were vortexed, and 3 mL of enzymatic reagent (Cholesterol Liquiform—76-2/100, LabTest, Lagoa Santa, MG, Brazil) was added. The solution was then kept in a water bath at 37 °C for another 10 min. After this period, samples were read using a spectrophotometer (Shimadzu UV-1800, Shimadzu Corporation, Kyoto, Japan) at λ = 500 nm to determine the cholesterol concentration in the samples.

### 2.4. Statistical Analysis

Individual rabbits were considered experimental units for the meat quality parameters. A completely randomized design was used in a 2 × 3 factorial scheme. Two independent variables, rigor mortis stage and carcass freezing time, were set as fixed effects. Two main effects and their interactions were tested.

The results were analyzed using Proc GLM in the SAS statistical software (version 9.1, SAS Institute Inc., 2002–2003, Cary, NC, USA). All data were subjected to analysis of variance (two-way ANOVA), and means were compared using Tukey’s test, with the significance level set at 5%.

## 3. Results

The parameters related to the color of Botucatu rabbit meat were significantly influenced by the rigor mortis stage and the carcasses’ freezing time. On the dorsal surface of the LL muscle, a greater (*p* = 0.035) red intensity and (*p* = 0.024) saturation index were recorded for the meat frozen post-rigor. A significant interaction was observed between the rigor mortis stage and carcass freezing time for luminosity (*L**) on the dorsal surface of the LL muscle (Table 1).

Meanwhile, on the ventral surface, the yellow intensity (*b**) was higher (*p* < 0.001) in the LL muscle of the carcasses frozen in the post-rigor phase, whereas the chroma saturation index showed the opposite trend to the dorsal surface, with higher (*p* = 0.023) values recorded for the carcasses frozen in the pre-rigor phase. A significant interaction was also observed between the rigor mortis stage and carcass freezing time for luminosity (*L**) on the ventral surface of the LL muscle, as detailed in Table 2.

Both the dorsal and ventral surfaces of the LL muscle showed higher levels (*p* < 0.05) of yellowness (*b**) as well as lower levels of redness (*a**) and chroma saturation (*C**) as the storage time progressed (Table 1 and Table 2). The luminosity of both surfaces of the LL muscle samples was lower (*p* < 0.05) at 12 months of freezing compared to the fresh samples, regardless of the rigor mortis phase at the freezing time. These color results may have been influenced, among other factors, by the significant fluctuations in pH values recorded during prolonged storage, as can be seen in Table 3.

There was an interaction (*p* < 0.05) between the rigor mortis phase and the carcass freezing time for the pH variable, the breakdown of which is presented in Table 3. During the first 6 months of storage, meat frozen in the pre-rigor phase proved to be less acidic compared to meat frozen in the post-rigor period, expressed by higher pH values. At 12 months of storage, there was no significant difference between the pH of samples frozen in the pre- and post-rigor phases.

Regarding the freezing time, the fresh samples (without freezing) showed higher muscle pH levels compared to the frozen samples, regardless of the freezing time elapsed. At 6 months, the post-rigor meat was more acidic, with the lowest pH among the freezing times, which may have affected the variables already presented, such as the CL, intensity of red (*a**), intensity of yellow (*b**), and luminosity (*L**).

This pH variation also appears to have influenced the cooking loss, which was more intense (*p* < 0.001) at 6 months of freezing, but not the water-holding capacity, which did not show statistical significance at any time.

The carcasses’ storage time also influenced (*p* < 0.001) the sarcomere length (SL), myofibrillar fragmentation index (MFI), and lipid oxidation (TBARS) levels. In the fresh meat, sarcomeres had a shorter length (1.80 µm) compared to those in the meat frozen for 6 and 12 months (2.14 and 2.02 µm, respectively). The MFI was higher in the post-rigor samples (67.40) than in the pre-rigor ones (52.08). Regarding the storage time, the MFI was higher in the carcasses frozen for 12 months, as well as the lipid oxidation levels, which increased gradually during freezing, reaching a maximum value at 12 months (0.281 → 0.809 mg MDA kg^−1^).

An interaction could also be observed between the rigor mortis phase at freezing and the freezing time for shear force (SF), with a breakdown shown in Table 4.

The fresh samples showed higher shear force values that decreased over the storage time, indicating the increased tenderness of LL muscle frozen for up to 12 months. Fresh samples at the pre-rigor stage still exhibited lower SF (34.23 N) values compared to the post-rigor meat (45.11 N) chilled for 24 h, indicating a greater tenderness. At other times, there was no significant differentiation in terms of SF values based on the rigor mortis stage at freezing.

Another interaction was verified for the percentage of moisture in the LL samples depending on the factors studied, as shown in Table 5. At 0 and 12 months of freezing, the percentage of moisture was higher for the pre-rigor meat, while at 6 months, the values were the same. At the end of the freezing storage period (12 months), both the pre- and post-rigor meat presented lower moisture contents compared to the other times.

Significant effects were also observed on the chemical composition of Botucatu rabbit meat, with higher percentages of protein, fat, and carbohydrate and a lower mineral content as the carcass storage time progressed, as demonstrated in Table 5. No significant differences were found for these variables based on the rigor mortis stage at the time of carcass freezing.

The gross energy of the meat (*p* < 0.001) increased over the frozen storage period (Table 6), likely due to the reduction in moisture content of the samples over the respective periods, causing other chemical components with known caloric value (fat, protein, and carbohydrate) to be more concentrated in the analyzed samples. The cholesterol concentration in the LL samples and the soluble protein present in the exudate released from the carcasses at thawing did not show significant differences in response to any of the factors studied.

## 4. Discussion

The pH level stands out as one of the most impactful chemical properties affecting meat quality. Its decline in the initial hours post-mortem directly influences other physicochemical properties of the muscle, such as color and tenderness [40]. In live rabbits, muscle pH levels are almost neutral, decreasing sharply in the early post-mortem stages and less markedly in subsequent hours [41,42]. This pH decline results from the accumulation of inorganic phosphoric acid due to the depletion of adenosine triphosphate (ATP) [43] and of muscle glycogen reserves leading to lactic acid formation [44], which is a critical process in the transformation of muscle into meat. However, cooling hot carcasses inhibits this sequence of reactions, delaying the decrease in pH level and the subsequent development of rigor mortis [45].

The pH values found in our study were consistent with those reported in the literature for rabbit meat. Jolley et al. [46] recorded initial (pH_i_) and ultimate (pH_24h_) pH values of 6.74 and 6.00 in *Longissimus thoracis et lumborum* (LTL), respectively. Meanwhile, Wang et al. [47] observed a pH_24h_ of 5.87 for Hyla rabbits.

pH values directly influence meat brightness, an important parameter that impacts sensory perception and plays a significant role in consumer acceptance, as it is closely linked to the perceived quality of the meat [48]. Thawed post-rigor meat exhibits a higher level of brightness in comparison to thawed pre-rigor meat, as supported by Claus and Sørheim [8], who noted that the outer surface of post-rigor beef had a greater level of luminosity than pre-rigor beef. As the pH approaches approximately 5.6, which is close to the isoelectric point of myofibrillar proteins like actin and myosin, these proteins tend to denature and aggregate. These structural changes alter proteins refractive indices, leading to increased light scattering and, consequently, a higher level of meat reflectivity [23]. This relationship was easily observed at six months of freezing, when the post-rigor carcasses, which exhibited the lowest pH value (5.56 ± 0.10), also showed the highest light intensities (L_dorsal_: 62.12 ± 0.52; L_ventral_: 60.90 ± 0.24) among all of the samples evaluated.

The decrease in pH observed during frozen storage can be related to the release of exudates that eventually inhibit the food’s buffering capacity, leading to the denaturation of myofibrillar proteins and release of H^+^ ions, resulting in an increase in solute concentration [49]. Senesi et al. [50], for instance, found that rabbit meat frozen for up to six months had a higher level of acidity (pH: 5.3–5.5) than fresh meat (pH: 5.9). Carrillo-Lopez et al. [51] recorded ultimate pH levels ranging from 5.68 to 5.72 at the end of frozen storage.

It is interesting to note that frozen meats in different stages of rigor mortis were only equal in terms of acidity at 12 months of freezing, when the pH values of the pre-rigor LL muscle no longer exceeded those of the post-rigor LL muscle. This is because freezing does not completely halt enzymatic reactions; thus, the maturation process continues even at subzero temperatures until all glycolytic reserves are depleted.

However, a slight increase in pH for the post-rigor group by the end of the evaluated period was observed in our study. This finding is consistent with observations by other authors who reported similar pH increases after chilling and subsequently freezing rabbit meat 24 h post-slaughter [5,52]. Similarly, Dalle Zotte et al. [53] in one of their studies recorded that the pH of frozen rabbit meat remained stable for up to three months, followed by a progressive rise until its 15th month of storage. The rise in pH of frozen meat may be attributed to the production of ammonia, amines, and other substances resulting from protein degradation by meat spoilage microorganisms and endogenous enzymes [54,55]. Conversely, additional protein deamination leading to the formation of high amounts of total volatile nitrogen [53] and the post-mortem proteolysis of myofibrillar proteins during rigor mortis resolution [56] could also have slightly increased the muscle pH levels. It is possible to infer that rigor mortis continued to develop throughout frozen storage, as well as during the 24 h of thawing preceding the physicochemical analyses, especially in the meat frozen without prior chilling, that is, in the pre-rigor stage, which may have impacted the evaluated parameters.

The luminosity of the pre-rigor meat was lower at 12 months of freezing, coinciding with lower pH values. Similar behavior was noted by Lan et al. [57], who recorded that the *L** of rabbit thigh meat decreased from 64.07 to 56.89 and 56.41 over freezing periods at −4 °C and −2.5 °C, respectively. Similarly, Dalle Zotte et al. [58] also observed reduced luminosity in the same cut after freezing, with *L** values decreasing from 62.3 to 58.2.

Associated with brightness, the most interesting color coordinate for the rabbit industry is the red intensity (*a**). Low positive values indicate a lower degree of redness, presenting a pinkish-red meat, which is more attractive to the modern consumer, who increasingly seeks the incorporation of white meats into their diet. Meat redness is influenced by the levels of myoglobin in the muscle tissue and the surrounding oxygen pressure. These factors contribute to color changes during refrigerated storage, during which the meat can transition from a brown color, due to metmyoglobin, to a bright red color, due to oxymyoglobin [44]. In rabbits, myoglobin levels are very low (0.02% in natural matter) compared to red meat species like cattle (0.39–0.48% in natural matter) [59]. Nevertheless, the dorsal surface of the LL muscle showed a greater red intensity after rigor mortis was established, as verified by Carrillo-Lopez et al. [51], who recorded increased rabbit meat redness as the cooling time increased in the early post-mortem hours. In post-mortem muscle, mitochondria remain active and influence meat color through oxygen consumption. Consequently, metmyoglobin expression is reduced, giving the surface exposed to air a more reddish hue [60], which, coupled with muscular acidification, lightens the muscle as rigor develops and the pH approaches the isoelectric point of muscle proteins.

The yellow intensity, on the other hand, was more pronounced on the ventral surface of the LL muscle in the samples that underwent rigor mortis. Similarly, Claus and Sørheim [8] found that the outer surface of pre-rigor meat showed a greater yellow intensity compared to post-rigor meat. This increase was equally observed for both surfaces over 12 months of freezing. Generally, while *b** increases, *a** and *L** tend to decrease during frozen storage [61,62]. Chwastowska-Siwiecka et al. [52] also observed changes in color parameters in response to freezing, with an increase in yellow intensity and a decrease in red intensity in the LTL muscle of frozen rabbits.

This reduction in the redness of rabbit meat due to the freezing time has been recently demonstrated in other studies [51,57]. Lan et al. [57] observed that the red intensity in the thigh muscles of rabbits decreased with the storage time, similar to what was observed in the present study for the loin muscles of Botucatu rabbits. Throughout the storage period, the activity of MetMb reductase (MRE), an enzyme that degrades metmyoglobin (MetMb) into deoxymyoglobin (DeoMb) and then oxygenates it back to oxymyoglobin (OxyMb), decreases as the meat is frozen [63]. Consequently, MetMb gradually accumulates on the meat surfaces, resulting in a lower red intensity [64].

The cooking loss was also influenced by the pH values associated with the storage time, demonstrating an inverse relationship between these parameters. Higher pH values are generally associated with reduced levels of cooking loss [4], which in turn affect meat juiciness and appearance [65]. Lan et al. [57] also found that the cooking loss in frozen rabbit meats gradually increased with the storage time and that freezing prevented, but did not inhibit, protein degradation. Thus, as muscle acidification progresses and pH values approach the isoelectric point of myofibrillar proteins, greater cooking losses are observed due to sarcoplasmic and myosinic denaturation by specific proteases and myofibrillar fragmentation [42].

The initial exposure to freezing temperatures is a critical factor in meat refrigeration [66]. Chilling carcasses prior to freezing allows the temperature of the meat to decrease gradually, resulting in a more uniform temperature distribution throughout the product. This controlled chilling process promotes the formation of smaller and more uniform ice crystals, thereby preserving muscle fibers and enhancing meat texture. The formation of ice crystals induces ultrastructural damage to myofibrils and concentrates meat solutes, leading to biochemical changes at the cellular level and alterations in frozen meat quality compared to fresh meat [49].

Freezing meat before rigor mortis can lead to thaw rigor, by which the formation of ice crystals disrupts the sarcoplasmic reticulum within muscle cells. This disruption can cause a sudden release of calcium into the sarcoplasm upon thawing, which leads to intense muscle contractions, a reduction in sarcomere length, and tougher meat [67]. However, in this study, these effects were not observed, likely due to the muscle cell rupture caused by the crystals formed during freezing storage, which increased the level of tenderness in thawed meat [68].

Increased tenderness, expressed by the shear force, was observed for both the pre- and post- rigor meat throughout the 12-month storage time. The findings by Lan et al. [57] for rabbit meat frozen for up to 36 days support our results by showing that in addition to the redness intensity, brightness, and muscle microstructure integrity, the shear force also decreased with the storage time, as reported by Vergara et al. [69] and Carrillo-Lopez et al. [51]. Although tenderness is one of the most crucial attributes for consumers, significantly influencing their perception of meat quality and contributing to market acceptance [70], the effects of combining chilled and frozen storage on meat quality remain relatively unexplored. This highlights the need for further research on this important issue for the meat industry and the cold chain.

The disruption of connective tissues and myofibrils, as evidenced by the increased myofibrillar fragmentation index—a reliable predictor of meat tenderness [71]—explains the significant decrease in shear force during freezing, resulting in the enhanced tenderness of the LL muscle. This tenderness is promoted not only by ice crystal formation [49] but also by enzymatic activity from bacterial and endogenous enzymes (calpains and cathepsins) that degrade key proteins such as troponin, connectin, and desmin [51,57]. Additionally, it is linked to greater muscle fiber relaxation, as evidenced by the increase in sarcomere length with an extended frozen storage time [23].

A greater degree of cellular tissue rupture and ice crystal formation between and within fibers results in increased water redistribution and solute accumulation during freezing [52,72]. Consequently, higher exudate losses occur during thawing. This leads to the concentration of hydrophobic compounds such as fat and other proteinaceous and energetic components in samples subjected to prolonged storage, in an inversely proportional relationship to moisture content [49,73]. Minerals are also lost along with the exuded liquid, proportional to the moisture loss. This increase in the concentration of major meat components results in higher gross energy values as the storage time increases. Fadlilah et al. [74] also reported a gross energy of 110.47 Kcal/100 g for White New Zealand rabbit meat, a breed globally recognized for its potential. These caloric values are close to those reported by the FAO (Food and Agriculture Organization of the United Nations), which specifies 120 Kcal/100 g for rabbit meat [75].

Rabbit meat’s proximate composition can vary due to various intrinsic and extrinsic factors affecting the animals; however, the values found in this study were consistent with those reported in the scientific literature [76,77]. This demonstrates the excellent nutritional quality of Botucatu rabbit meat, characterized by low levels of fat and cholesterol, a reduced caloric content, and significant amounts of protein and minerals.

Bianospino et al. [78] also reported similar average levels of 21.54% protein, 76.11% moisture, 0.69% fat, and 1.24% ash for Botucatu rabbits, confirming the nutritional quality of LL muscle in the evaluated lineage. Chilled carcasses show higher water losses due to evaporation during air chilling, resulting in a higher moisture content in pre-rigor LL muscle (with chilling) compared to post-rigor LL muscle (with 24 h of chilling), similar to observations in beef carcasses, which can lose up to 2% of their initial weight during chilling due to evaporative losses [79].

Despite the observed increase in fat content, prolonged freezing did not cease but inhibited lipid oxidation, one of the main factors contributing to meat quality deterioration [80]. Lipid oxidation produces hydroperoxides, which degrade into secondary products such as ketones, aldehydes, and other carbonyl compounds responsible for the development of rancid and pungent flavors in meat [47,81]. Among these, malonaldehyde (MDA), a secondary product of lipid oxidation, has been widely used as an indicator of lipid oxidation in TBARS assays. When its levels exceed the recommended limit, it may lead to quality loss and consumer rejection.

The gradual increase in TBARS values during the freezing of rabbit meat was expected due to its relatively high content of polyunsaturated fatty acids [76], making it susceptible to lipid oxidation during extended storage periods [82], as previously observed by other authors [4,47,49]. Lan et al. [57] also observed a continuous increase in TBARS from 0.05 mg MDA/kg in fresh rabbit meat to 1.037 and 0.773 mg MDA/kg when frozen for 36 days at −2.5 °C and −4 °C, respectively. However, throughout the 12-month storage period, prolonged freezing did not cause the TBARS values to exceed 2 mg MDA/kg of meat, which is considered the threshold for this analysis [83], thus ensuring meat quality. Additionally, alongside the increased formation of thiobarbituric acid reactive substances, the decrease in *L** and *a** values suggests that lipid oxidation contributed to accelerated myoglobin oxidation and consequently to the discoloration of frozen rabbit meat [47,49].

## 5. Conclusions

The rigor mortis phase affects the meat color in Botucatu rabbits, resulting in increased redness on the dorsal surface and yellowness on the ventral surface of post-rigor *Longissimus lumborum* muscle. Moreover, pre-rigor fresh meat is softer, moister, and less acidic compared to meat that undergoes rigor mortis after 24 h of chilling.

Long-term freezing tenderizes the meat, whether it is frozen pre- or post-rigor mortis, preserving its physical, chemical, and nutritional quality for up to 12 months. Minor changes in color, acidity, and lipid oxidation, as well as losses in moisture and minerals may occur over this period.

## Figures and Tables

**Table 1 animals-14-02510-t001:** Breakdown (±SEM) of the interaction between rigor mortis phase and carcass freezing time for luminosity (*L**) on the dorsal surface of the *Longissimus lumborum* (LL) muscle in Botucatu rabbits. Means (±SEM) of red intensity (*a**), yellow intensity (*b**), and chroma saturation index (*C**) values on the dorsal surface of the LL muscle. *p*-values are presented.

Dorsal Surface							
Rigor Mortis Phase (n = 10)	Carcass Freezing Time (n = 10)			*p*-Value			CV (%)
	0 Month	6 Months	12 Months				
	*L** *Dorsal*			*p* (RP)	*p* (CFT)	*p* Int. (RP × CFT)	
Pre-rigor	52.06 ± 0.54 ^Aa^	51.21 ± 0.61 ^Bab^	50.62 ± 0.14 ^Bb^	<0.001	<0.001	<0.001	5.64
Post-rigor	53.14 ± 0.32 ^Ab^	62.12 ± 0.52 ^Aa^	53.11 ± 0.33 ^Ab^				
		** *a* ** *****		** *b* ** *****		** *C* ** *****	
*Rigor mortis phase (RP)*							
Pre-rigor		6.78 ± 0.17 ^B^		1.68 ± 0.13		6.99 ± 0.34 ^B^	
Post-rigor		7.07 ± 0.03 ^A^		1.46 ± 0.12		7.22 ± 0.85 ^A^	
*Carcass freezing time (CFT)*							
0 month		7.25 ± 0.31 ^A^		0.40 ± 0.15 ^B^		7.26 ± 0.80 ^A^	
6 months		6.42 ± 0.32 ^B^		1.80 ± 0.17 ^AB^		6.67 ± 0.14 ^AB^	
12 months		6.35 ± 0.54 ^B^		2.51± 0.23 ^A^		6.83 ± 0.21 ^B^	
*p-value*							
*p* (RP)		0.035		0.542		0.024	
*p* (CFT)		<0.001		0.014		<0.001	
*P* (RP × CFT)		0.572		0.312		0.131	
CV (%)		20.18		11.98		5.61	

Means followed by different letters in columns (uppercase ^A,B^) and rows (lowercase ^a,b^) differ significantly according to Tukey’s test (*p* < 0.05). SEM: standard error of the mean; *a**: redness intensity; *b**: yellowness intensity; *C**: chroma saturation; *L**: luminosity; RP: rigor mortis phase; CFT: carcass freezing time; Int: interaction; and CV: coefficient of variation.

**Table 2 animals-14-02510-t002:** Breakdown (±SEM) of the interaction between rigor mortis phase and carcass freezing time for luminosity (*L**) on the ventral surface of the *Longissimus lumborum* (LL) muscle in Botucatu rabbits. Means (± SEM) of redness intensity (*a**), yellowness intensity (*b**), and chroma saturation index (*C**) values on the ventral surface of the LL muscle. *p*-values are presented.

Ventral Surface							
Rigor Mortis Phase (n = 10)	Carcass Freezing Time (n = 10)			*p*-Value			CV (%)
	0 Month	6 Months	12 Months				
	*L** *Ventral*			*p* (RP)	*p* (CFT)	*p* Int. (RP × CFT)	
Pre-rigor	53.12 ± 0.47 ^Aa^	52.83 ± 0.35 ^Bab^	49.95 ± 0.99 ^Bb^	<0.001	<0.001	<0.001	6.76
Post-rigor	53.14 ± 0.66 ^Aab^	60.90 ± 0.24 ^Aa^	52.09 ± 0.44 ^Ab^				
		***a****		***b****		***C****	
*Rigor mortis phase (RP)*							
Pre-rigor		7.03 ± 0.03		0.76 ± 0.13 ^B^		7.07 ± 0.21 ^A^	
Post-rigor		6.93 ± 0.17		1.93 ± 0.12 ^A^		6.81 ± 0.32 ^B^	
*Carcass freezing time (CFT)*							
0 month		7.92 ± 0.32 ^A^		0.43 ± 0.15 ^C^		7.94 ± 0.46 ^A^	
6 months		6.35 ± 0.54 ^B^		1.15± 0.23 ^B^		6.45 ± 0.28 ^B^	
12 months		6.05 ± 0.31 ^B^		2.46 ± 0.17 ^A^		6.53 ± 0.50 ^B^	
*p-value*							
*p* (RP)		0.448		<0.001		0.023	
*p* (CFT)		<0.001		<0.001		<0.001	
*p* Int. (RP × CFT)		0.062		0.129		0.098	
CV (%)		24.47		13.39		0.14	

Means followed by different letters in columns (uppercase ^A–C^) and rows (lowercase ^a,b^) differ significantly according to Tukey’s test (*p* < 0.05). SEM: standard error of the mean; *a**: redness intensity; *b**: yellowness intensity; *C**: chroma saturation; *L**: luminosity; RP: rigor mortis phase; CFT: carcass freezing time; Int: interaction; and CV: coefficient of variation.

**Table 3 animals-14-02510-t003:** Breakdown (±SEM) of the interaction between rigor mortis phase and carcass freezing time for pH level of the *Longissimus lumborum* (LL) muscle in Botucatu rabbits. Means (±SEM) of pH values, cooking loss (CL), and water-holding capacity (WHC) of the LL muscle. *p*-values are presented.

Rigor Mortis Phase (n = 10)	Carcass Freezing Time (n = 10)			*p*-Value			CV (%)
	0 Month	6 Months	12 Months				
pH				*p* (RP)	*p* (CFT)	*p* Int. (RP × CFT)	
Pre-rigor	6.86 ± 0.14 ^Aa^	5.77 ± 0.11 ^Ab^	5.80 ± 0.12 ^Ab^	<0.001	<0.001	<0.001	1.77
Post-rigor	5.91 ± 0.11 ^Ba^	5.56 ± 0.10 ^Bc^	5.86 ± 0.13 ^Ab^				
		**CL (%)**			**WHC (%)**		
*Rigor mortis phase (RP)*							
Pre-rigor		27.95 ± 0.42			66.08 ± 0.36		
Post-rigor		29.37 ± 0.32			64.28 ± 0.43		
*Carcass freezing time (CFT)*							
0 month		27.49 ± 0.91 ^B^			64.42 ± 1.36		
6 months		33.91 ± 1.01 ^A^			65.68 ± 1.08		
12 months		30.63 ± 0.60 ^AB^			65.32 ± 0.76		
*p-value*							
*p* (RP)		0.076			0.355		
*p* (CFT)		<0.001			0.201		
*p* Int. (RP × CFT)		0.161			0.083		
CV (%)		31.75			6.17		

Means followed by different letters in columns (uppercase ^A,B^) and rows (lowercase ^a–c^) differ significantly according to Tukey’s test (*p* < 0.05). SEM: standard error of the mean; CL: cooking loss; WHC: water-holding capacity; RP: rigor mortis phase; CFT: carcass freezing time; Int: interaction; and CV: coefficient of variation.

**Table 4 animals-14-02510-t004:** Breakdown (±SEM) of the interaction between the rigor mortis phase and carcass freezing time for shear force of the *Longissimus lumborum* (LL) muscle in Botucatu rabbits. Means (±SEM) of shear force (SF), sarcomere length (SL), myofibrillar fragmentation index (MFI), and lipid oxidation (TBARS) values in the LL muscle. *p*-values are presented.

Rigor Mortis Phase (n = 10)	Carcass Freezing Time (n = 10)			*p*-Value			CV (%)
	0 Month	6 Months	12 Months				
*Shear Force (N)*				*p* (RP)	*p* (CFT)	*p* Int. (RP × CFT)	
Pre-rigor	34.23 ± 1.89 ^Ba^	21.57 ± 1.11 ^Aab^	12.75 ± 1.02 ^Ab^	0.13	<0.001	<0.001	52.92
Post-rigor	45.11 ± 1.69 ^Aa^	18.63 ± 1.30 ^Ab^	12.16 ± 1.09 ^Ab^				
		**SL** **(µm)**		**MFI**		**TBARS** **(mg MDA kg^−1^)**	
*Rigor mortis phase (RP)*							
Pre-rigor		2.05 ± 0.07		52.08 ± 1.39 ^B^		0.463 ± 0.02	
Post-rigor		1.92 ± 0.02		67.40 ± 1.14 ^A^		0.541 ± 0.01	
*Carcass freezing time (CFT)*							
0 month		1.80 ± 0.06 ^B^		51.21 ± 3.27 ^B^		0.281 ± 0.05 ^C^	
6 months		2.14 ± 0.04 ^A^		50.30 ± 1.73 ^B^		0.415 ± 0.05 ^B^	
12 months		2.02 ± 0.04 ^AB^		77.70 ± 3.17 ^A^		0.809 ± 0.08 ^A^	
*p-value*							
*p* (RP)		0.216		<0.001		0.311	
*p* (CFT)		<0.001		<0.001		0.01	
*p* Int. (RP × CFT)		0.463		0.327		0.064	
CV (%)		13.84		13.84		19.12	

Means followed by different letters in columns (uppercase ^A,B,C^) and rows (lowercase ^a,b^) differ significantly according to Tukey’s test (*p* < 0.05). SEM: standard error of the mean; SF: shear force; SL: sarcomere length; MFI: myofibrillar fragmentation index; TBARS: lipid oxidation; RP: rigor mortis phase; CFT: carcass freezing time; Int: interaction; and CV: coefficient of variation.

**Table 5 animals-14-02510-t005:** Breakdown (±SEM) of the interaction between the rigor mortis phase and carcass freezing time for the moisture content of the *Longissimus lumborum* muscle in Botucatu rabbits and means of chemical composition (±SEM). *p*-values are presented.

Rigor Mortis Phase (n = 10)	Carcass Freezing Time (n = 10)			*p*-Value			CV (%)
	0 Month	6 Months	12 Months				
	*Moisture (%)*			*p* (RP)	*p* (CFT)	*p* Int. (RP × CFT)	
Pre-rigor	74.08 ± 0.36 ^Aa^	73.45 ± 0.41 ^Aab^	73.10 ± 0.34 ^Ab^	0.426	<0.001	<0.001	2.31
Post-rigor	73.45 ± 0.41 ^Ba^	73.41 ± 0.46 ^Aa^	72.75 ± 0.37 ^Bb^				
	**Protein (%)**	**Mineral Content (%)**		**Fat (%)**		**Carbohydrate (%)**	
*Rigor mortis phase (RP)*							
Pre-rigor	22.92 ± 1.36	1.28 ± 0.08		2.05 ± 0.13		0.21 ± 0.06	
Post-rigor	22.76 ± 1.36	1.24 ± 0.06		2.06 ± 0.13		0.24 ± 0.07	
*Carcass freezing time (CFT)*							
0 month	22.63 ± 1.36 ^B^	1.31 ± 0.16 ^A^		1.96 ± 0.13 ^B^		0.33 ± 0.07 ^B^	
6 months	22.74 ± 1.36 ^B^	1.27 ± 0.08 ^AB^		2.03 ± 0.13 ^B^		0.53 ± 0.06 ^A^	
12 months	23.16 ± 1.36 ^A^	1.16 ± 0.18 ^B^		2.20 ± 0.13 ^A^		0.55 ± 0.03 ^A^	
*p-value*							
*p* (RP)	0.477	0.069		0.158		0.355	
*p* (CFT)	<0.001	<0.001		0.015		0.012	
*p* Int. (RP × CFT)	0.25	0.201		0.429		0.131	
CV (%)	12.47	1.98		1.56		8.98	

Means followed by different letters in columns (uppercase ^A,B^) and rows (lowercase ^a,b^) differ significantly according to Tukey’s test (*p* < 0.05). SEM: standard error of the mean; RP: rigor mortis phase; CFT: carcass freezing time; Int: interaction; and CV: coefficient of variation.

**Table 6 animals-14-02510-t006:** Means of gross energy, cholesterol, and soluble protein (±SEM) of the *Longissimus lumborum* muscle of Botucatu rabbits as a function of rigor mortis phase and carcass freezing time.

	Gross Energy (Kcal/100 g)	Cholesterol (mg/100 g)	Soluble Protein (mg/mL)
*Rigor mortis phase (RP)*			
Pre-rigor	110.97 ± 1.16	62.36 ± 1.03	0.124 ± 0.03
Post-rigor	110.54 ± 1.14	62.60 ± 1.23	0.135 ± 0.03
*Carcass freezing time (CFT)*			
0 month	109.48 ± 1.06 ^B^	63.60 ± 1.16	-
6 months	111.35 ± 1.31 ^AB^	63.46 ± 1.16	0.129 ± 0.03
12 months	114.64 ± 1.26 ^A^	60.37 ± 1.16	0.130 ± 0.03
*p*-value			
*p* (RP)	0.831	0.068	0.832
*p* (CFT)	<0.001	0.212	0.067
*p* Int. (RP × CFT)	0.312	0.234	0.067
CV (%)	8.32	11.02	6.31

Means followed by different letters in columns 
(uppercase ^A,B^) differ significantly according to Tukey’s test (*p* < 0.05). SEM: standard error of the mean; RP: rigor mortis phase; CFT: carcass freezing time; Int: interaction; and CV: coefficient of variation.

## Data Availability

The data that support the findings will be available in Repositorio Institucional da UNESP at http://hdl.handle.net/11449/244719 (accessed on 1 June 2024) following a delay from the date of publication to allow time for the article to be published first.

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
