# Peer review of "Influence of Long-Term Freezing of Carcasses in Pre- and Post-Rigor Mortis Stages on the Technological and Nutritional Parameters of the Longissimus lumborum Muscle of Botucatu Rabbits"

_animals, 2024, doi:10.3390/ani14172510_

Round 1

Reviewer 1 Report

Comments and Suggestions for Authors

The manuscript titled “Influence of long-term freezing of carcasses in pre- and post-rigor mortis stages on the technological and nutritional parameters of the longissimus lumborum muscle of Botucatu rabbits” presents a well-designed study that explores an important aspect of meat science. The experimental design is appropriate for the objectives of the study, and the analytical techniques used are relevant and properly executed. The results are clearly presented, and the discussion is well-written, providing a thorough interpretation of the findings in the context of existing literature.

The topic addressed is of significant interest to the readers of this journal, particularly those involved in meat science and animal production. The findings contribute valuable insights into the effects of long-term freezing on the quality of rabbit meat, which is of importance for both producers and consumers. Before publicacion, some improvements should be made:

  1. Alignment of Conclusion with Abstract: It is recommended that the conclusions drawn in the manuscript be carefully aligned with the abstract. Consistency between these sections will help reinforce the key messages of the study and ensure that readers gain a clear understanding of the main findings and their implications.
  2. Details on Rabbit Rearing and Diet: While the study focuses on the effects of freezing at different stages of rigor mortis, it would be beneficial to include more information about the rearing conditions of the rabbits. Specifically, the manuscript would be enhanced by detailing the type of diet provided to the rabbits and the final weights they achieved at the end of the production period. This information could help readers better understand the baseline conditions that may influence the muscle characteristics under investigation.
  3. Carcass Yield Data: The manuscript would also benefit from the inclusion of data on carcass yield, particularly concerning any differences observed between carcasses frozen before and after rigor mortis. If such data is available, it should be presented to give a more comprehensive overview of the impact of freezing at different stages on overall meat production efficiency.

Reviewer 2 Report

Comments and Suggestions for Authors

The study titled “Influence of long-term freezing of carcasses in pre- and post-rigor mortis stages on the technological and nutritional parameters of the longissimus lumborum muscle of Botucatu rabbits” examined the effects of prolonged freezing on the quality of Botucatu rabbit meat, both before and after rigor mortis. The findings revealed that extended freezing time improves the meat’s tenderness, irrespective of the rigor stage at the time of freezing, while maintaining its physical, chemical, and nutritional integrity, with only slight variations in color and composition. There are few questions should be addressed:

1. The author declare that This aligns with recommendations from the United States Department of Agriculture, which states that rabbit carcasses can be frozen for up to 12 months without compromising quality. However, the entire experiment period of this study is 12 months. So, it is hard to distinguish the result after 12 month.

2. In the part of Simple summary, the author should briefly introduce the background of this research.

3. The format of the references needs to be standardized and uniform.

4. Some of the references are too outdated. The author should update them with the research within latest 5 years.

Reviewer 3 Report

Comments and Suggestions for Authors

This research aims to assess the impact of long-term storage on the quality of frozen Botucatu rabbit meat in pre- and post-rigor stages by analyzing the stability of technological and nutritional parameters of the longissimus lumborum (LL) over months. The topic of this research is interesting and valuable. In view of the scientific significance and application value of the paper, and the existing problems can be solved by further modification and improvement, it is suggested to give the opportunity to revise.

1.     The abstract is a bit messy. Line 34-35 “Lightness…with storage” whose lightness, red intensity, and saturation decreased ?  Whose yellow intensity increased ? 

2.     In terms of experimental design, sensory evaluation was lacking to assess the effect of long-term freezing on rabbit meat quality under two different methods.

3.     In terms of experimental design more, authors aim to analyze the effect of pre- and post-rigor on the frozen meat quality. Therefore, authors set two groups, one group without chilling, and another one is chilled under temperature of 2 ± 2ºC for 24 hours, in order to finish the post-rigor. And then the meats were frozen at -20 ºC. There is one question, in order to reduce the ice crystal in meat when they were frozen, the chilling treatment is advised which can help meat to decrease temperature, to reduce the ice crystal. So please explain the effect of pre-cooling on meat quality as compared with post-rigor.

4.     Different thawing methods may have a significant effect on the quality of frozen meat, the specific thawing method should be indicated.

5.     Results: This section mentions color, PH, cooking loss, shear force, muscle fiber breakage index, lipid oxidation, chemical composition, crude energy, cholesterol, and soluble protein, but does not correspond to a detailed description of how these influencing factors affect meat quality.

6.     Discussion: While the effects of pH changes on meat color and water retention are discussed, the molecular mechanisms behind these changes can be further explored, such as the denaturation of specific proteins and how they affect the structure and function of meat. The discussion can suggest directions for future research to address issues or limitations not yet addressed in current research.

7.     Line 124, “ longissimus lumborum (LL) ”, as mentioned in the abstract section, you just need to use LL here, the latter is also. Such as 144, 159, 164

8.     Line 219 , “P=0.0345 , P=0.0241”  et al. the ‘P’ here should be italicized.

9.     Line 246-247 , “Both surfaces......at 12 months of freezing”. There is no decline in Post-rigor at 6 months. Moreover, the results should be analyzed based on significant difference.

10.  Line 265 , “ fresh samples ” does not agree with Line 294 , “samples in natura” .

11.  All the tables in the article should use three-wire tables. Or reorganize them to make them more clearly.

12.  Line 253 and Table 4, the expression of PPC and CRA in the table is inconsistent with the expression of CL and WHC mentioned in the Table 4 title.

13.  Line 282 and Table 6, the MFI should be modified to FMI consistent with above, consistency should be expressed.

14.  Line 340, pHi means ?.

15.  Line 343, the relationship between pH and meat brightness was not clear. Line 263, "After 12 months of storage, there was no significant difference between the pH of samples frozen in the pre- and post-rigor phases"is there consistency with the following analysis?

Comments on the Quality of English Language

it can be improved after English language revision.

Round 2

Reviewer 3 Report

Comments and Suggestions for Authors

Authors have revised according the reviewer suggestions.